# Role of TRPC3 in Right Ventricular Dilatation under Chronic Intermittent Hypoxia in 129/SvEv Mice

**DOI:** 10.3390/ijms241411284

**Published:** 2023-07-10

**Authors:** Do-Yang Park, Woon Heo, Miran Kang, Taeyoung Ahn, DoHyeon Kim, Ayeon Choi, Lutz Birnbaumer, Hyung-Ju Cho, Joo Young Kim

**Affiliations:** 1Department of Otolaryngology, Ajou University School of Medicine, Suwon 16499, Republic of Korea; entdyp@ajou.ac.kr; 2Department of Pharmacology, Yonsei University College of Medicine, Seoul 03722, Republic of Korea; woon.heo@dong-wha.com (W.H.); taeyoungahn@yuhs.ac (T.A.); dhk9313@yuhs.ac (D.K.); aychoi@yuhs.ac (A.C.); 3Department of Otorhinolaryngology, Yonsei University College of Medicine, Seoul 03722, Republic of Korea; mrkang1123@yuhs.ac; 4Laboratory of Signal Transduction, National Institute of Environmental Health Sciences, Research Triangle Park, Durham, NC 27709, USA; birnbau1@gmail.com; 5Institute of Biomedical Research (BIOMED), Catholic University of Argentina, Buenos Aires C1107AFF, Argentina; 6The Airway Mucus Institute, Yonsei University College of Medicine, Seoul 03722, Republic of Korea

**Keywords:** obstructive sleep apnea, chronic intermittent hypoxia, transient receptor potential cation channel 3, right ventricle, right ventricle hypertrophy, right ventricle dilation, endothelin

## Abstract

Patients with obstructive sleep apnea (OSA) exhibit a high prevalence of pulmonary hypertension and right ventricular (RV) hypertrophy. However, the exact molecule responsible for the pathogenesis remains unknown. Given the resistance to RV dilation observed in transient receptor potential canonical 3*(Trpc3)^−/−^* mice during a pulmonary hypertension model induced by phenylephrine (PE), we hypothesized that TRPC3 also plays a role in chronic intermittent hypoxia (CIH) conditions, which lead to RV dilation and dysfunction. To test this, we established an OSA mouse model using 8- to 12-week-old 129/SvEv wild-type and *Trpc3^−/−^* mice in a customized breeding chamber that simulated sleep and oxygen cycles. Functional parameters of the RV were evaluated through analysis of cardiac cine magnetic resonance images, while histopathological examinations were conducted on cardiomyocytes and pulmonary vessels. Following exposure to 4 weeks of CIH, *Trpc3^−/−^* mice exhibited significant RV dysfunction, characterized by decreased ejection fraction, increased end-diastole RV wall thickness, and elevated expression of pathological cardiac markers. In addition, reactive oxygen species (ROS) signaling and the endothelin system were markedly increased solely in the hearts of CIH-exposed *Trpc3^−/−^* mice. Notably, no significant differences in pulmonary vessel thickness or the endothelin system were observed in the lungs of wild-type (WT) and *Trpc3^−/−^* mice subjected to 4 weeks of CIH. In conclusion, our findings suggest that TRPC3 serves as a regulator of RV resistance in response to pressure from the pulmonary vasculature, as evidenced by the high susceptibility to RV dilation in *Trpc3^−/−^* mice without notable changes in pulmonary vasculature under CIH conditions.

## 1. Introduction

OSA is a common breathing disorder during sleep that poses a significant public health concern, affecting approximately 3–9% of the general population [1]. Accumulating evidence suggests that OSA is associated with various health issues such as uncontrolled hypertension, ventricular dysfunction, coronary artery disease, and rhythm disorders, including an increased risk of sudden cardiac death [2,3]. In patients with OSA, pulmonary hypertension (PH) can lead to right ventricular hypertrophy (RVH) and dysfunction [4].

During episodes of apnea and hypoventilation in OSA, minor fluctuations in intrathoracic pressure occur, which can affect the relaxation phase of the heart (diastole). This can result in atrial dilatation, arrhythmias, and pulmonary congestion [5]. These processes can trigger excessive pulmonary vasoconstriction and structural changes in the blood vessels, including endothelial damage, endothelial cell proliferation, intimal invasion of myofibroblast-like cells, and increased fibrosis [6]. Consequently, these changes contribute to RVH and increased pressure in the RV, while also significantly reducing the compliance of the pulmonary artery (PA) [7]. 

Initially, RVH caused by increased pressure in the RV is a compensatory response, but it can progress to RV failure. Some patients, despite having similar RV afterloads and masses, develop adaptive RVH, which is associated with improved functional capacity and survival. On the other hand, others develop maladaptive RV hypertrophy, characterized by dilatation, fibrosis, and RV failure [8]. Alterations in the structure and function of RV have been shown to predict clinical outcomes in OSA-related cardiopulmonary disease [9]. However, there is a lack of consistent evidence regarding the functional and morphological changes in the RV caused by OSA, as well as the factors responsible for controlling these changes.

Endothelin-1 (ET-1), produced by endothelial cells, plays a role in inducing vasoconstriction and cell proliferation in the smooth muscle cells of the pulmonary artery, thereby contributing to the pathogenesis of pulmonary arterial hypertension (PAH) [10]. Additionally, patients with PAH exhibit an upregulation of the RV myocardial endothelin axis [11], which may be a compensatory mechanism to counteract increased afterload by enhancing contractility and cardiac output. ET-1 exerts its effects on smooth muscle through the endothelin type A (ETAR) and type B (ETBR) receptors. ETAR activation leads to proliferation via mitogen-activated protein kinase and tyrosine kinase pathways, while ETBR exhibits anti-apoptotic properties in endothelial cells [12].

Increased pulmonary workload resulting from respiratory distress and inadequate oxygen supply during cardiac activity leads to cardiac dilatation through RV hypertrophy and contributes to the development of atherosclerosis, primarily because of elevated pulmonary artery pressure (PAP) and oxygen-related stress [8]. TRPC3 channels have been suggested to facilitate this process [13]. TRPC3 is involved in cardiac dilation, and its regulation is influenced by molecules associated with hypoxia [14]. Additionally, TRPC3 plays a significant role in cardiac hypertrophy, implicating cardiac function and pathology through the calcineurin–nuclear factor of an activated T-cell (NFAT) signaling pathway [15,16]. Overexpression of the TRPC3 in cardiomyocyte increases the entry of calcium ions through the store-operated channel, which is crucial for the myocardial hypertrophic response to neurostimulatory factors or excessive blood pressure [17]. In our previous study, *Trpc3^−/−^* mice demonstrated resistance to phenylephrine-induced pathological cardiac hypertrophy, particularly in the RV [18]. Furthermore, TRPC3 plays a pivotal role in the release of nitric oxide (NO), a molecule involved in vessel relaxation following contraction [19,20]. It is well established that patients with OSA [21,22] experienced impaired endothelium-dependent vasodilation, reduced NO release [23], and diminished expression of endothelial nitric oxide synthase under the condition of chronic hypoxia, resulting in decreased vasodilation [24]. However, the precise role of TRPC3 in the pulmonary vasculature and heart under CIH, which simulates the hypoxia/reoxygenation cycles seen in OSA, remains unclear.

In this study, our objective is to investigate the effects and mechanisms of TRPC3 on CIH-induced changes in the RV by utilizing an OSA mouse model with both WT and *Trpc3^−/−^* mice. We will assess cardiac function using cine magnetic resonance imaging (MRI) and explore potential associations with ROS and the endothelin system, comparing these factors between WT and *Trpc3^−/−^* 129/SvEv mice.

## 2. Results

### 2.1. Cardiac Function Measured by MRI

Cardiac function of WT and *Trpc3^−/−^* mice was assessed using cine-MRI under sham condition and CIH conditions. The MRI scans were conducted in either the axial or longitudinal axis, following the parameters described in the Methods section (Figure 1A,B, Appendix A). While the systolic blood pressures of *Trpc3^−/−^* and WT mice did not show significant changes because of CIH, we observed that WT mice exhibited a recovery in systolic blood pressure after a sudden decrease three weeks into the CIH protocol (Appendix A). As shown in Figure 1C, the mean ejection fraction (EF) of the RV in the *Trpc3^−/−^* CIH group was significantly reduced. In addition, the RV cavity space during the contraction and release phases appeared larger in the *Trpc3^−/−^* CIH group compared to the WT CIH group (Figure 1A,B). Furthermore, there was a significant increase in the thickness of RV wall during the end-diastolic (ED) phase, but not the end-systolic (ES) phase, exclusively in the *Trpc3^−/−^* group (Figure 1D,E). To verify the absence of *Trpc3* expression in the hearts of *Trpc3^−/−^* mice, we conducted Western blotting along with RT-PCR analysis and along with the expression of *Trpc1* and *Trpc6* (Figure 1F,G). These analyses confirmed that TRPC3 expression was indeed absent in *Trpc3^−/−^* cells. Based on these findings, it can be concluded that CIH leads to a decline in RV function accompanied by an increased in RV wall thickness, specifically in *Trpc3^−/−^* mice.

### 2.2. TRPC3 Deletion and Hypoxia Aggravated RV Function

The histology analysis of WT and *Trpc3^−/−^* mice hearts using H&E staining did not reveal significant differences (Figure 2A). However, the parenchyma of RV (Figure 2B) was notably reduced, and the interstitial fibrosis level of RV (Figure 2C) was significantly increased only in *Trpc3^−/−^* mice. Furthermore, the mRNA levels of the cardio-pathological markers (atrial natriuretic peptide (*Anp),* brain natriuretic peptide *(Bnp)*, and β myosin heavy chain (*β-Mhc*) were significantly elevated in the heart tissues of the *Trpc3^−/−^* CIH group compared to the WT sham, WT CIH, and *Trpc3^−/−^* sham groups (Figure 2D). Examining the levels of ROS signaling molecules (*p22Phox*, *Nox2*, and *Nox4*) and the hypoxia marker (*Hif-1α*) in the heart samples, we observed a significant increase exclusively in *Trpc3^−/−^* mice within the CIH groups (Figure 2E). Both mRNA and protein expression of *Et-1* in the heart samples were significantly elevated in both the WT and *Trpc3^−/−^* CIH groups (Figure 2F), with a more pronounced increase observed in the *Trpc3^−/−^* CIH group. Interestingly, *Et-2* mRNA expression in the heart samples showed a significant increase solely in the *Trpc3^−/−^* CIH group. Additionally, while *EtbR* mRNA expression in the heart samples increased in both the WT and *Trpc3^−/−^* CIH groups, *Trpc3^−/−^* mice exhibited an approximately sevenfold increase (Figure 2G). These results indicate that *Trpc3^−/−^* mice experienced CHI-induced pathologic changes in the heart, accompanied by elevated level of inducers such as ROS and *Hif-1α*, along with alterations in components of the endothelin system.

### 2.3. Changes in RV Function Were Not Influenced by Lung Vascularity

The PA wall thickness (Figure 3B) and muscularization area (Figure 3C) in both CIH groups (WT CIH and *Trpc3^−/−^* CIH) were significantly increased compared to the sham groups (WT sham and *Trpc3^−/−^* sham), as shown in Figure 3A–C. However, the degree of increase caused by CIH was similar between WT and *Trpc3^−/−^* mice. The mean levels of ROS (*p22Phox*, *Nox2*, and *Nox4*) in the lungs were slightly higher in the *Trpc3^−/−^* groups (*Trpc3^−/−^* sham and *Trpc3^−/−^* CIH) compared to in the WT groups (WT sham and WT CIH), but these differences were not statistically significant according to Figure 3D. Interestingly, the CIH-induced increase in *Hif-1α* expression observed in *Trpc3^−/−^* mice was not observed in the lung tissue. When comparing the mRNA levels of *Et-1*, *Et-2*, and ET receptors (*EtaR* and *EtbR*) in the lung samples, no significant differences were found among the WT CIH, Trpc3^−/−^ sham, and *Trpc3^−/−^* CIH groups compared to the WT sham group. However, the mean *Et-3* mRNA level in the *Trpc3^−/−^* group was higher than in the other groups, although the difference was not statistically significant (Figure 3E). These findings indicate that the lungs of *Trpc3^−/−^* mice experienced histological changes and vascularization because of CIH treatment, similar to WT mice. However, there were no noticeable changes in the ROS or ET system in the lungs, further supporting the idea that the functional changes in the RV were specifically associated with *Trpc3^−/−^* mice.

## 3. Discussion

In this study using an animal model of obstructive sleep apnea (OSA), we compared the changes in PA (pulmonary artery) between WT and *Trpc3^−/−^* mice and found no significant differences. However, *Trpc3^−/−^* mice exhibited pronounced RV dysfunction caused by CIH, as evidenced by Figure 1C and Figure 2B,C. We also observed an increase in the activity of the endothelin axis in *Trpc3^−/−^* mice (Figure 2E,F). Furthermore, only *Trpc3^−/−^* mice showed an increase in end-diastolic RV wall thickness, a decrease in RV function, elevated levels of markers associated with cardiac pathology, increased signaling molecules related to ROS, and a substantial increase in endothelin-1 (ET-1) and ET-2 (Figure 1 and Figure 2). These findings suggest that *Trpc3^−/−^* experienced an abnormal pathological process in the RV in response to CIH, probably because of the loss of normal response to PAH resulting from TRPC3 deletion.

Patients with PAH have been reported to experience impaired coronary systolic blood flow and right ventricular ischemia [25]. Compensatory RV hypertrophy occurs in response to increased resistance in the pulmonary circulation [26]. However, sustained increase in resistance can lead to progressive consequences such as systolic dysfunction, decompensation, dilatation, and ultimately, RV failure [7]. The increased RV pressure in turn increases wall stress and myocardial oxygen demand while interfering with myocardial perfusion and triggering an exaggerated inflammatory response [6].

The abnormal pathological process observed exclusively in the RV of *Trpc3^−/−^* mice under CIH conditions emphasizes the essential protective role of TRPC3 against RV failure. Previous studies using the same *Trpc3^−/−^* mice have reported TRPC3 as an important factor in cardiac hypertrophy, particularly in the LV [18,27,28]. While significant changes in LV were not detected under the conditions of this study, the decreased RV function (Figure 1E), increased end-systolic RV wall thickness, significant reduction in cardiomyocyte parenchyma, and increased interstitial fibrosis (Figure 2B,C) were exclusively observed in *Trpc3^−/−^* mice, indicating uncompensated RV failure. The regulatory processes involved in the transition from compensatory RV hypertrophy to uncompensated RV failure in PAH are currently limited in our understanding. Our study suggests that the regulation of cardiac hypertrophy by TRPC3 plays an important role in this conversion process.

Furthermore, under chronic hypoxia, there is a decrease in NO release and endothelial nitric oxide synthase expression, leading to reduced vasodilation [23,24]. Considering the critical role of TRPC3 in NO release following vessel constriction, the sustained pressure applied to the RV in *Trpc3^−/−^* mice may result from prolonged pulmonary vasoconstriction [19,20]. Although *Trpc3* deficiency did not contribute to pulmonary vascular remodeling, the reduced NO release may have contributed to the RV overload observed in *Trpc3^−/−^* mice.

In previous experiments, we found that *Trpc3^−/−^* mice were resistant to RV dilation induced by high-dose (PE stimulation, which was observed in WT mice [18]. PE, an α1 agonist, increased mesenteric arterial pressure and elevated pulmonary arterial pressure when administrated in high doses, resulting in impaired RV systolic function [29,30]. This phenomenon was confirmed again by our study (Appendix A). The reduced RV dilation in *Trpc3^−/−^* mice during PE infusion can be attributed to decreased PA contraction caused by the absence of TRPC3. A similar contraction tendency induced by PE in the mesenteric artery and pulmonary artery can be assumed based on previous studies [31]. The absence of RV dilation in *Trpc3^−/−^* mice under PE infusion may be attributed to a decrease in pulmonary artery contraction because of the absence of TRPC3, suggesting that the loss of TRPC3’s hypertrophic function in the myocardium was not problematic. On the other hand, under CIH conditions that induce PH, *Trpc3^−/−^* mice exhibited decreased RV function (Figure 1C), a significant decrease in RV parenchyma (Figure 2B), and increased fibrosis (Figure 2C), as well as increased expression of pathological markers (Figure 2D). The severe RV dilation observed in *Trpc3^−/−^* mice under CIH conditions might be a result of the absence of the cellular hypertrophy necessary to resist the increase in PAP induced by CIH, even if there may have been some decrease in contractility because of the absence of TRPC3. There is a possibility that the specific hypoxic conditions of CIH, when met with the absence of TRPC3, further exacerbated the progression of appropriate compensatory processes. 

In studies on LV remodeling, increased activity of *Nox2/p22phox* leads to excessive generation of ROS, causing pathological remodeling, fibrosis, and functional abnormalities of the myocardium through signaling pathways associated with inflammation, hypertrophy, and cell death, ultimately contributing to abnormal changes in the LV [32]. It has been reported that the expression of NOX2 protein and ROS production is not dependent on mRNA levels but rather regulated by the inhibition of NOX2 protein degradation through direct interaction with TRPC3 protein [33,34]. From this perspective, the increases in *p22phox* and *Nox2* mRNA expression levels in *Trpc3*-deficient RV may serve as compensatory mechanisms to adapt to hypoxic stress. However, further investigation is needed to understand the precise relationship between TRPC3 deficiency, hypoxic stress, and upregulation of *Nox2/p22phox* within the RV and the complex mechanisms, such as adaptation induced by hypoxia in the heart. Additionally, exploring downstream signaling pathways is essential to better understand the role of TRPC3 in RV resistance to pulmonary pressure. Key molecules that require further research include G-protein-coupled receptors (GPCRs), phosphoinositide 3-kinase (PI3K) and Akt, calcium-dependent pathways, protein kinase C (PKC) isoforms, and mitogen-activated protein kinases (MAPKs) [35]. Investigating the activation of these molecules and their effects can provide valuable insights into the interaction with TRPC3 and the resistance of the RV.

It is very surprising that there was no significant difference in PA in this study. However, it was probably due to the vascular characteristics of the 129S strain mouse used in this study. In a previous report examining the major vascular stiffness indices of frequently used inbred mice, the blood pressure and heart rate of healthy animals were the same in all strains. However, a distinct difference in vascular stiffness indices was reported [36]. That is, C57Bl/6J mice, which were mainly used in pulmonary-artery-related studies, showed the lowest tensile stiffness and the highest acetylcholine-induced vascular relaxation, whereas 129S mice showed high tensile stiffness and the lowest acetylcholine-induced vascular relaxation. If pulmonary vasoconstriction and relaxation can be measured functionally in the CIH situation, it will be possible to determine whether there is no significant difference in pulmonary vascular changes based on these vascular characteristics. Studies investigating CIH utilizing the 129S mouse strain might more distinctly delineate the impact on cardiac functioning as opposed to vascular effects. The severe RV dysfunction observed in this study is likely attributable to an RV-specific mechanism in response to CIH.

Endothelin, the most potent vasoconstrictor in the cardiovascular system, acts on pulmonary artery smooth muscle cells (PASMC) to induce vasoconstriction and cell proliferation, thereby contributing to the pathogenesis of PAH [10]. Patients with PAH exhibit increased levels of endothelin-1 (ET-1) in their bloodstream [29], which correlates with pulmonary vascular resistance and right atrial pressure [37]. ET-1 functions through the endothelin A receptor (ETAR) and endothelin B receptor (ETBR). ETAR is expressed in smooth muscle and causes vasoconstriction, while ETBR extensively acts on the endothelium and promotes vasodilation [38]. ET-1 and ETBR are associated with various signaling pathways, including Bcl2, epidermal growth factor receptor (EGF-R), and mitogen-activated protein kinase (PK) cascade, facilitating the survival and hypertrophy of cardiomyocytes under pressure overload [10]. Considering the vasodilatory and proliferative effects of *EtbR*, the increased expression of *ErbR* in *Trpc3^−/−^* hearts under conditions of CIH may be a compensatory response of the heart to pulmonary hypertension resulting from impaired NO secretion [10]. Under CIH conditions, *Trpc3^−/−^* hearts exhibited a 2-fold increase in ET-1 levels and a 17-fold increase in ET-2 expression compared to WT hearts, which did not show CIH-induced changes (Figure 2F). ET-1 upregulates HIF-1α, which can promote maladaptive remodeling and anaerobic metabolism [39]. It is not clear whether TRPC3 directly inhibits the expression of ET-2 or if the absence of TRPC3 leads to increased ET-2 expression. However, the significant increase in *Et-2* expression observed only in *Trpc3^−/−^* mice under CIH conditions can further constrict the pulmonary artery, contributing to an abnormal progression in the right ventricle and creating a repetitive cycle of maladaptive changes.

Our study has limitations. First, we were unable to observe the progression of RV dysfunction in response to CIH, nor did we investigate the influence of the stimulus duration on the outcomes. Evaluating the interaction between PAP and the RV and PA is crucial to assess the severity and progression of PH. However, investigations into the functional outcomes associated with these factors need to be more comprehensive. Further studies are required to elucidate the molecular mechanisms through which TRPC3 contributes to RV adaptation with PH. The small sample size that was due to the low pregnancy rate of the *Trpc3^−/−^* mice we used, the potential confounding factors, and the inability to directly generalize the findings from animal to human OSA constitute significant limitations of this study. Future research should incorporate such functional evaluations to comprehensively investigate the impacts of TRPC3 deficiency on PH and the associated functional outcomes.

To the best of our knowledge, no studies have been conducted to investigate the impact of OSA and CIH on the molecular mechanisms affecting RV function; specifically, there have been no studies focusing on TRPC3 in this context. Previous reports have mentioned OSA-induced hypertension, left ventricular hypertrophy, dilation, and heart failure, but these studies were primarily limited to the LV. Our study is significant as it identified the important role of TRPC3 in maintaining RV function under CIH conditions.

In conclusion, TRPC3 deficiency resulted in severe RV dysfunction after 4 weeks of CIH exposure, without noticeable phenotypic changes in the lungs. These findings suggested that *Trpc3* serves as a regulator of RV resistance in response to pulmonary vasculature pressure.

## 4. Material and Methods

### 4.1. Experimental Animals and CIH Exposure

The Trpc3^−/−^ 129/SvEv mice were described in a previous study [35]. *Trpc3^−/−^* (n = 13) and WT (n = 12) mice were used in this study. All animals analyzed in this study were maintained according to the Yonsei Medical Center animal research requirements, and all procedures were approved by the Committee on Animal Research at the Yonsei Medical Center (protocol number 2017-0035). All experimental procedures were reviewed and approved by the Institutional Animal Research Ethics Committee at the Yonsei Medical Center (Seoul, South Korea) and performed in accordance with relevant guidelines and regulations, including ARRIVE guidelines.

*Trpc3^−/−^* and WT (129/SvEv) mice were acclimated to the study conditions for 1 week and randomly divided into sham (*Trpc3^−/−^* sham and WT sham) and CIH (*Trpc3^−/−^* CIH and WT CIH) groups. Customized intermittent hypoxia chambers were used in the CIH group (*Trpc3^−/−^* CIH and WT CIH) to create the CIH mouse model. The animals were housed in standard breeding cages and provided with standard mouse chow and water in a temperature- and light-controlled room (23–25 °C, 12 h each of daylight and darkness). During CIH exposure, the mice were transferred from breeding cages to a custom-built CIH chamber coupled with a gas-control delivery system. Gas-control delivery equipment (Live Cell Instrument, Namyangju, Korea) was built to regulate the nitrogen and oxygen flow into the customized chamber. The CIH groups were subjected to intermittent hypoxia of 5–21% nadir ambient oxygen every 2 min, by using nitrogen infusion into the daytime chambers. Adult mice have an inverted sleep–wake cycle relative to humans by remaining asleep during the day and awake at night. Thus, CIH was administered for 12 h during the day for 8 weeks.

### 4.2. Measurement of Cardiac Function by MRI

Cardiac MRI was performed with a 9.4T magnet (Bruker BioSpin MRI, Ettlingen, Germany) using an electrocardiography (ECG)-triggered fast low-angle shot (FLASH) gradient echo with cine sequence. Briefly, the mice were anesthetized in a hyphenated chamber using isoflurane (1.5% isoflurane/98.5% O_2_) and supinely placed in a cradle. To optimize cardiac MRI, which can create a strong motion artifact, an ECG electrode (SA Instruments, Inc., Stony Brook, NY, USA) was inserted into the mouse forepaws, and a respiration loop was taped across the chest for R-wave detection. After the mice were positioned in the MRI chamber with the heart at the center and scouted for short- and long-axis heart orientation using a double-gated segmented gradient echo sequence, shimming and pulse calibration were automatically performed using ParaVision 5.0. The ECG-triggered experiments were conducted by the application of a FLASH cine sequence with the following imaging parameters: echo time = 2.1 ms; repetition time = 6.0 ms; field of view = 30.0 × 30.0 mm^2^; matrix size = 256 × 256 pixels; excitation pulse = 20°; and slice thickness = 0.5 mm.

Cross-sectional views of the left ventricle (LV) and RV were visualized using an MRI microimaging system, comprising a magnet. Inspection of the cine loop revealed that the frames representing the largest and smallest cross-sectional LV and RV m-mode images represent the systolic and diastolic phases of the heart, respectively. Standard echocardiography included the assessment of LV and RV fractional shortening (FS), myocardial masses, and internal diameters during systole and diastole. The LV and RV internal diameters were traced and measured using the built-in software of the equipment.

The cross-sections of the end-systolic and -diastolic RV areas were used to modify the calculation method for measuring the stroke volume of the right ventricle. Five cross-sectional areas were collected to generate a cone shape, and the RV volume was the sum of four truncated cone volumes. Because the right ventricle is crescent shaped and distinct from that of the left ventricle, we determined its volume as 1/3 of the truncated cone volume, as per Equation (1).
(1)$$V=\pi {r_{1}}^{2}h+\pi {r_{n}}^{2}h+\sum_{i=1}^{4}\frac{\pi h}{9}\left({r_{i}}^{2}+r_{i}r_{i+1}+{r_{i+1}}^{2}\right)$$

### 4.3. Specimen Harvesting and Histological Analysis

The mice were sacrificed after blood collection. The thorax was opened, and the heart and lungs were carefully harvested. The right lung specimen was inflated by tracheal infusion of low-melting-point agarose at an airway pressure of 25 cmH_2_O and immersed in 4% paraformaldehyde (PFA) for 48 h. Heart and left lung specimens were frozen in liquid nitrogen and subsequently stored at −80 °C.

### 4.4. RV Cardiacmyocyte Area Analysis

Immediately after echocardiography imaging, the mice were sacrificed, and blood was collected under isoflurane anesthesia (3:1 O_2_). Hearts were immediately frozen in liquid nitrogen (for molecular analysis) or immersion-fixed in 4% PFA for paraffin embedding and histomorphometry analysis. Transverse cross-sections of the RV were paraffin-embedded, and 5 µm sections were stained with hematoxylin and eosin (H&E) to measure the cardiomyocyte surface area. Masson’s trichrome staining was used to measure the fibrosis level of each sample. For all histological studies, three pictures were randomly taken from the myocardium of the RV. Cardiomyocyte size was evaluated by measuring the cell perimeter and surface area in those with a visible nucleus. Masson’s trichrome staining level was calculated by (blue intensity of RV × blue area of RV)/cardiomyocyte parenchyma of RV. Image J 1.36b (http://rsbweb.nih.gov/ij/ (accessed on 21 June 2023)) was used for stereological analysis and pixel quantification.

### 4.5. Pulmonary Vascular Morphology Analysis

The right lung was fixed in 4% formalin for 48 h, transferred to 30% sucrose/phosphate-buffered saline, and stored overnight at 4 °C. After the lung was embedded in the optimal cutting temperature compound, it was frozen at −80 °C, and cut into 10 μm thick slices using a cryostat. To visualize the fibrotic muscularized region around the PA wall, H&E-stained slides were stained with Masson’s trichrome and were imaged using an inverted light microscope (IX73-F22PH, Olympus, Shinjuku Monolith, Japan); the dimensions of PA walls in the images were calculated using the two-dimensional region tool of MetaMorph (Molecular Devices, San Jose, USA). The value was converted to real μm^2^ based on the square area by using a bar (50 μm). More than 10 wall positions from 10 arteries were measured, and the averages were summed. 

### 4.6. Measuring mRNA Expression in the Heart and Lungs

Heart and left lung samples were stored at −80 °C and subjected to reverse transcription–polymerase chain reaction analysis. Total RNA was isolated using TRIzol reagent (Invitrogen, Life Technologies, Carlsbad, CA, USA) following the manufacturer’s protocol. RNA was quantified using NanoDrop, and absorbance ratios at 260/280 nm were >1.80 for all samples studied. One microgram of total RNA was reverse transcribed in a 20 μg reaction mixture containing cDNA Synthesis Master Mix, under conditions indicated by the manufacturer (Agilent Technologies, Santa Clara, CA, USA). 

A quantitative real-time polymerase chain reaction was performed using a CFX Connect Real-Time PCR Detection System (Bio-Rad Laboratories, Inc., Hercules, CA, USA) and TOPreal qPCR 2× PreMix (SYBR Green with high ROX; Enzynomics, Daejeon, Korea), according to the manufacturers’ instructions. RNA levels in the heart and lung samples were measured using ROS markers (*p22Phox*, *Nox2*, and *Nox4*), intermittent hypoxia marker (*Hif-1α*), and endothelin markers (*Et-1, Et-2, Et-3, EtγA*, and *EtγB*). The RNA levels in the heart samples were additional measures of cardio-pathological markers (*Anp, Bnp*, and *β-Mhc*). The primers used are listed in Table 1.

### 4.7. Enzyme-Linked Immunosorbent Assay Analysis

Blood samples were obtained from the tail vein of each mouse. Serum was obtained by centrifugation and stored at −20 °C. Serum ET levels were measured using an ET-1 ELISA kit (ADI-900-020A; Enzo Life Sciences, Farmingdale, NY, USA).

### 4.8. Statistical Analysis

Statistical analysis was performed using two-way analysis of variance with a multiple-comparisons follow-up test, using GraphPad Prism version 7.00 (GraphPad Software, San Diego, CA, USA) unless otherwise stated. Independent data of separate experiments are expressed as mean ± standard error. Statistical significance was set at *p* < 0.05.

## Figures and Tables

**Figure 1 ijms-24-11284-f001:**
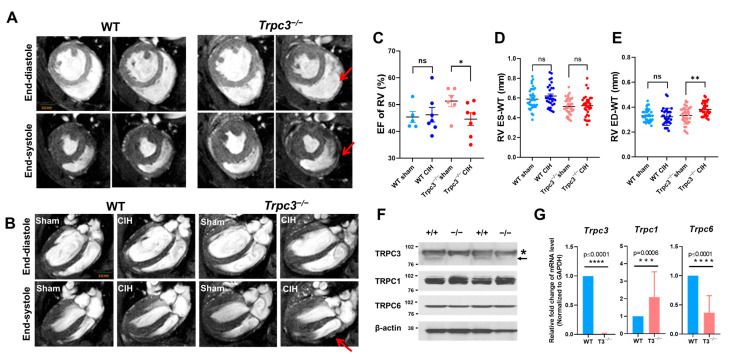
Representative cardiac long- and short-axis magnetic resonance images. (**A**) Axial end-diastole (ED) and end-systole (ES) phase views. The enlarged right ventricular (RV) cavity is indicated by red arrows. (**B**) Longitudinal view of the ED and ES phases. (All images were taken with the same magnitude.) (**C**) RV ejection fraction (RVEF). % Ejection fraction (EF) was calculated as follows: EF = (SV/EDV) × 100. The mean RVEF was significantly lower in the *Trpc3^−/−^* chronic intermittent hypoxia (CIH) group (n = 6 for each group). RV wall thickness of end-systole (**D**) and end-diastole (**E**) states. Protein expression (**F**) and mRNA expression (**G**) of *Trpc3* along with *Trpc1* and *Trpc6* in WT and *Trpc3^−/−^* hearts. Arrow indicates TRPC3 proteins band and asterisk indicates non-specific bands. Data are displayed as mean ± standard error (SE). Student’s multiple unpaired *t*-test was used to determine statistical significance (* *p* < 0.05, ** *p* < 0.01, and ns: not significant).

**Figure 2 ijms-24-11284-f002:**
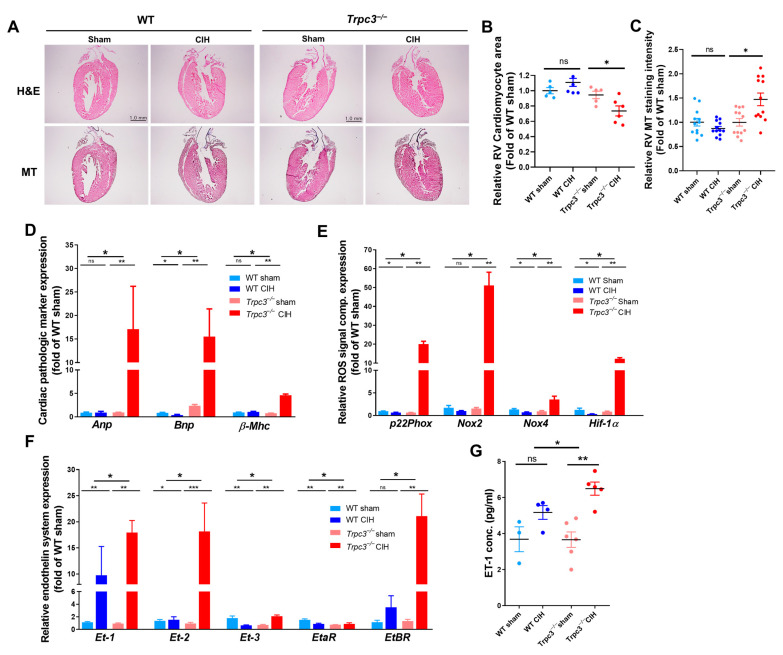
Effects of TRPC3 deficiency and CIH on cardiac hypertrophy. (**A**) Histological analysis of heart by hematoxylin and eosin (H&E) and Masson’s trichrome staining. Comparison of right ventricular cardiomyocyte area (**B**) and collagen deposition (**C**) of right ventricular cardiomyocyte area. (**D**) Quantitative comparison of mRNA expression of *Anp*, *Bnp*, and *β-Mhc*) from the heart tissue. *Trpc3^−/−^* mice under CIH displayed significant overexpression of *Anp, Bnp*, and *β-Mhc* compared to the other groups. Data are expressed as mean ± SE. (**E**) The mRNA expression of reactive oxygen species-related markers (*p22Phox, Nox2,* and *Nox4*) and hypoxia marker (*Hif-1α*) significantly increased in the heart tissue of *Trpc3^−/−^* mice with CIH. (**F**) The differential expression pattern of the endothelin (ET) system in the WT and *Trpc3^−/−^* mice treated with CIH. (**G**) Comparison of serum ET-1 protein levels. The mean serum level of ET-1 was also significantly higher in both WT and *Trpc3^−/−^* mice treated with CIH than that of the sham groups. Data are expressed as mean ± SE. (* *p* < 0.05, ** *p* < 0.01, *** *p* < 0.001, and ns; not significant).

**Figure 3 ijms-24-11284-f003:**
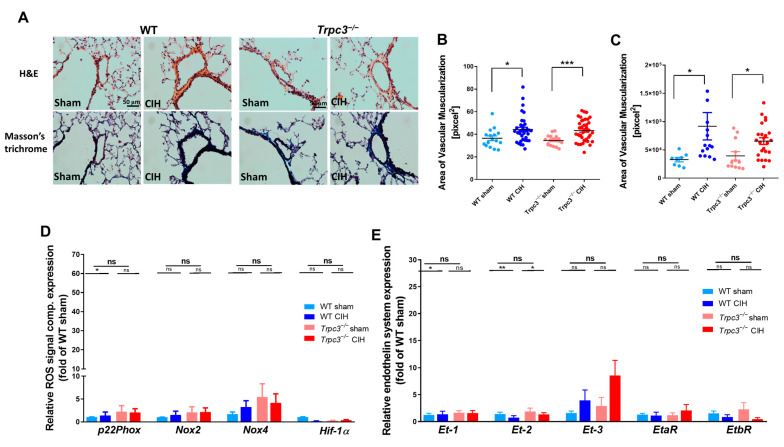
Effects of *Trpc3* deficiency and CIH on the lung phenotype. (**A**) Histologic evaluation with H&E and Masson’s trichrome staining showed increased vascular wall thickness in WT and *Trpc3^−/−^* mice under CIH exposure. (**B**) Comparison of vascular wall thickness and length increased in the WT and *Trpc3^−/−^* mice under CIH exposure. (**C**) Comparison of vascular muscularization area revealed increases in the WT and *Trpc3^−/−^* mice under CIH exposure. (**D**) The mRNA expression of reactive oxygen species-related markers (*p22Phox*, *Nox2*, and *Nox4*) and hypoxia marker (*Hif-1α*) in the lung. Data are expressed as mean ± SE (n = 3). (**E**) The expression pattern of the endothelin system in the WT and *Trpc3^−/−^* mice, with CIH exposure in the lung. Data are expressed as mean ± SE (n = 3). (* *p* < 0.05, ** *p* < 0.01, *** *p* < 0.001, and ns; not significant).

**Table 1 ijms-24-11284-t001:** Primer sequences for mRNA detection.

Gene	Forward Primer	Reverse Primer
*Trpc3*	5′-GCCATTGCCAGTGTGATCTA-3′	5′-AGGTTGGAGGCACCATCAA-3′
*Trpc1*	5′-GAATCGCGTAACCAGCTCAG-3′	5′-AGTGGGCCCAAAATAGAGCT-3′
*Trpc6*	5′-TCTCGAGTTGGGGATGCTTT-3′	5′-GCGAGAATGATTGGGGTCAC-3′
*p22Phox*	5′- GCCATTGCCAGTGTGATCTA -3′	5′-AGGTTGGAGGCACCATCAA-3′
*Nox2*	5′-TTCCAGTGCGTGTTGCTCGAC-3′	5′-GATGGCGGTGTGCAGTGCTAT-3′
*Nox4*	5′-GGA TCA CAG AAG GTC CCT AGC AG-3′	5′-GCG GCT ACA TGC ACA CCT GAG AA-3′
*Hif-1* *α*	5′-CAGTACAGGATGCTTGCCAAAA-3′	5′-ATACCACTTACAACATAATTCACACACACA-3′
*Et-1*	5′-CTGCTGTTC GTGACTTTCCA-3′	5′-AGCTCCGGTGCTGAGTTC-3′
*Et-2*	5′-TGCGTTTT CGTCGATGCTC-3′	5′-CTGTCTGTCCCGCAGTGTTCA-3′
*Et-3*	5′-TGGAC ACGCTTGCGTTGTACT-3′	5′-CGGAATAACTGGTGACATCTCTGG-3′
*EtγA*	5′-GAGGCGTAATGGCTGACAAT-3′	5′-GTGGTGCCCAGAAAGTTGAT-3′
*EtγB*	5′-CTCTGTTGGCTTCCCCTTC-3′	5′-CGATTGGATTGATGCAGGA-3′
*Anp*	5′-CCTCGTCTTGGCCTTTTGGCT-3′	5′-CCTCCAGGTGGTCTAGCAGGTTC-3′
*Bnp*	5′-AAGTCCTAGCCAGTCTCCAGA-3′	5′-CTGCCTTGAGACCGAAGG-3′
*β-Mhc*	5′-AGATGTTTTTGTGCCCGATGACA-3′	5′-CACCGTCTTGCCATTCTCCGT-3′

## Data Availability

The DOI for the data in this dataset is https://doi.org/10.6084/m9.figshare.22793945.

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
