# Peer review of "Role of TRPC3 in Right Ventricular Dilatation under Chronic Intermittent Hypoxia in 129/SvEv Mice"

_ijms, 2023, doi:10.3390/ijms241411284_

Round 1

Reviewer 1 Report

This study is an interesting finding that demonstrates the involvement of TRPC3 gene in the progression of right ventricular dysfunction caused by intermittent sleep hypoxia using TRPC3-deficient mice. The fact that the phenotype of right ventricular hypertrophy was nicely reproduced in an artificial experimental system using a hypoxic chamber, and the finding that hypertrophy was markedly suppressed in TRPC3-deficient mice is highly commendable. However, there are two major concerns with this study in terms of mechanistic analyses.

Major comments

1.     Figure 1F. anti-mouse TRPC protein antibodies are unreliable, making it difficult to trust the cut-out WB bands. The authors had better to show the results of mRNA expressions using qPCR analyses together and then claim that the results are similar to WB results.

2.     Figure 2D. It has been reported that in left ventricular myocardial remodeling, Nox2 protein expression and ROS production are not dependent on the mRNA level, but on the inhibition of Nox2 degradation through direct interaction with TRPC3 protein (ref., doi: 10.1038/srep37001., doi: 10.1172/jci.insight.93358.). Considering these reports, the increases in p22phox and Nox2 mRNA expression levels of TRPC3-deficient RVs might be a compensatory mechanism to adapt from hypoxic stress. The authors should either investigate whether Nox2 protein levels and ROS production are actually increased substantially, or at least cite the above references and discuss the differences with the mechanism of Nox2/p22phox participations in left ventricular remodeling.

Minor comment

1.     Abstract line 39. ‘form’ may be ‘from’.

None.

Author Response

My coauthors and I greatly appreciate the reviewer’s questions and comments, and we respond to them as below. We highlighted sentences that we changed or added in revised manuscript and Figures.

Reviewer 1.

Major comments

  1. Figure 1F. anti-mouse TRPC protein antibodies are unreliable, making it difficult to trust the cut-out WB bands. The authors had better to show the results of mRNA expressions using qPCR analyses together and then claim that the results are similar to WB results.

Response: I appreciate your comment to enhance the accuracy of our data. Following your advice, we have supplemented the revised manuscript with Figure 1G, which compares the expression levels at the mRNA level using Real-time PCR. The absence of a specific and reliable antibody with excellent performance for TRPC3, free from nonspecific bands, remains a challenge for many TRPC researchers. The confirmation of mRNA expression significantly complements and solidifies the representation of actual TRPC3 expression levels by the minor TRPC3 band.

  1. Figure 2D. It has been reported that in left ventricular myocardial remodeling, Nox2 protein expression and ROS production are not dependent on the mRNA level, but on the inhibition of Nox2 degradation through direct interaction with TRPC3 protein (ref., doi: 10.1038/srep37001., doi: 10.1172/jci.insight.93358.). Considering these reports, the increases in p22phox and Nox2 mRNA expression levels of TRPC3-deficient RVs might be a compensatory mechanism to adapt from hypoxic stress. The authors should either investigate whether Nox2 protein levels and ROS production are actually increased substantially, or at least cite the above references and discuss the differences with the mechanism of Nox2/p22phox participations in left ventricular remodeling.

Response: Thank you for your critical comments. While it is evident that TRPC3 directly interacts with Nox2 and influences its degradation, relying solely on mRNA expression for assessment falls short of reaching a comprehensive conclusion. Following your advice, we have incorporated the following statement along with the referenced papers in the discussion section. This addition will significantly assist readers in making more informed judgments based on established facts.

“In studies on LV remodeling, increased activity of Nox2/p22phox leads to excessive generation of ROS, causing pathological remodeling, fibrosis, and functional abnormalities of the myocardium through signaling pathways associated with inflammation, hypertrophy, and cell death, ultimately contributing to abnormal changes in the LV [35]. It has been reported that the expression of Nox2 protein and ROS production is not dependent on mRNA levels but rather regulated by the inhibition of Nox2 protein degradation through direct interaction with TRPC3 protein [36, 37]. From this perspective, the increases in p22phox and Nox2 mRNA expression levels in TRPC3-deficient RV may serve as compensatory mechanisms to adapt to hypoxic stress. However, further investigation is needed to understand the precise relationship between TRPC3 deficiency, hypoxic stress, and upregulation of Nox2/p22phox within the RV and the complex mechanisms, such as adaptation induced by hypoxia in the heart.”

Reference

  1. Murdoch, C. E.; Zhang, M.; Cave, A. C.; Shah, A. M., NADPH oxidase-dependent redox signalling in cardiac hypertrophy, remodelling and failure. Cardiovascular research 2006, 71, (2), 208-215.
  2. Kitajima, N.; Numaga-Tomita, T.; Watanabe, M.; Kuroda, T.; Nishimura, A.; Miyano, K.; Yasuda, S.; Kuwahara, K.; Sato, Y.; Ide, T.; Birnbaumer, L.; Sumimoto, H.; Mori, Y.; Nishida, M., TRPC3 positively regulates reactive oxygen species driving maladaptive cardiac remodeling. Scientific reports 2016, 6, (1), 37001.
  3. Shimauchi, T.; Numaga-Tomita, T.; Ito, T.; Nishimura, A.; Matsukane, R.; Oda, S.; Hoka, S.; Ide, T.; Koitabashi, N.; Uchida, K.; Sumimoto, H.; Mori, Y.; Nishida, M., TRPC3-Nox2 complex mediates doxorubicin-induced myocardial atrophy. JCI insight 2017, 2, (15).

Reviewer 2 Report

Overall, this research provides valuable insights into the role of TRPC3 in the pathogenesis of right ventricular dysfunction and pulmonary hypertension in obstructive sleep apnea (OSA). The study design and experimental approach are commendable, particularly the use of a mouse model of OSA and the comprehensive assessment of RV functional parameters and histopathological features.

There are some comments for the improvement as the following:

1. While the study suggests that TRPC3 is involved in RV resistance to pressure from the pulmonary vasculature, more detailed mechanistic investigations are needed to understand the exact molecular pathways involved. This could involve exploring downstream signaling pathways or potential interactions with other molecules implicated in RV dysfunction.

2. While the histological differences between wild-type (WT) and Trpc3−/− mice hearts are mentioned, additional details on the histopathological findings would strengthen the study. Describing specific changes in cardiomyocyte morphology or interstitial fibrosis could provide a better understanding of the structural alterations observed in Trpc3−/− mice.

3. The lack of significant changes observed in pulmonary vessel thickness and the endothelin system in the lungs of CIH-exposed mice is intriguing. However, to better understand the role of TRPC3 in pulmonary hypertension, it would be valuable to further investigate other aspects of pulmonary vessel remodeling, such as endothelial dysfunction or smooth muscle cell proliferation.

4. While the study focuses on RV functional parameters, it would be beneficial to also assess functional outcomes related to pulmonary hypertension, such as pulmonary arterial pressure measurements or right ventricular-pulmonary artery coupling. This would provide a more comprehensive understanding of the impact of TRPC3 deficiency on overall cardiovascular function.

5. It would be helpful for the authors to acknowledge and discuss any limitations of the study, such as sample size, potential confounding factors, or the generalizability of findings to human OSA. Addressing these limitations will strengthen the study's conclusions and provide context for future research directions.

1. Improve sentence structure: Several sentences in the discussion are long and complex, making it difficult for readers to follow the main points. Consider breaking down lengthy sentences into shorter, more concise ones to enhance clarity and readability.

2. Proofread for grammar and punctuation: The text contains some grammatical errors, such as incorrect verb tenses and missing articles. Carefully proofread the paper to correct these errors and ensure that the punctuation is accurate and consistent.

Author Response

My coauthors and I greatly appreciate the reviewer’s questions and comments, and we respond to them as below. We highlighted sentences that we changed or added in revised manuscript and Figures.

Reviewer 2.

Major comments

  1. While the study suggests that TRPC3 is involved in RV resistance to pressure from the pulmonary vasculature, more detailed mechanistic investigations are needed to understand the exact molecular pathways involved. This could involve exploring downstream signaling pathways or potential interactions with other molecules implicated in RV dysfunction.

 Response: Thank you for your feedback. In response to the comments from both reviewers, we have incorporated a discussion in the revised version that sheds light on the mechanistic aspects allowing for inference of the role of TRPC3 in the right ventricle and pulmonary artery (PA). We discuss the interaction between TRPC3 and Nox2 and delve into the diverse downstream signaling processes of TRPC3. We believe that these additions will not only enhance the scientific aspect of the manuscript but also contribute to the broader implications of our research. The additional content is outlined as follows:

“In studies on LV remodeling, increased activity of Nox2/p22phox leads to excessive generation of ROS, causing pathological remodeling, fibrosis, and functional abnormalities of the myocardium through signaling pathways associated with inflammation, hypertrophy, and cell death, ultimately contributing to abnormal changes in the LV [35]. It has been reported that the expression of Nox2 protein and ROS production is not dependent on mRNA levels but rather regulated by the inhibition of Nox2 protein degradation through direct interaction with TRPC3 protein [36, 37]. From this perspective, the increases in p22phox and Nox2 mRNA expression levels in TRPC3-deficient RV may serve as compensatory mechanisms to adapt to hypoxic stress. However, further investigation is needed to understand the precise relationship between TRPC3 deficiency, hypoxic stress, and upregulation of Nox2/p22phox within the RV and the complex mechanisms, such as adaptation induced by hypoxia in the heart. Additionally, exploring downstream signaling pathways is essential to understand better the role of TRPC3 in RV resistance to pulmonary pressure. Key molecules that require further research include G-protein-coupled receptors (GPCRs), phosphoinositide 3-kinase (PI3K) and Akt, calcium-dependent pathways, protein kinase C (PKC) isoforms, and mitogen-activated protein kinases (MAPKs) [37]. Investigating the activation of these molecules and their effects can provide valuable insights into the interaction with TRPC3 and the resistance of the RV.”

Reference

  1. Murdoch, C. E.; Zhang, M.; Cave, A. C.; Shah, A. M., NADPH oxidase-dependent redox signalling in cardiac hypertrophy, remodelling and failure. Cardiovascular research 2006, 71, (2), 208-215.
  2. Kitajima, N.; Numaga-Tomita, T.; Watanabe, M.; Kuroda, T.; Nishimura, A.; Miyano, K.; Yasuda, S.; Kuwahara, K.; Sato, Y.; Ide, T.; Birnbaumer, L.; Sumimoto, H.; Mori, Y.; Nishida, M., TRPC3 positively regulates reactive oxygen species driving maladaptive cardiac remodeling. Scientific reports 2016, 6, (1), 37001.
  3. Shimauchi, T.; Numaga-Tomita, T.; Ito, T.; Nishimura, A.; Matsukane, R.; Oda, S.; Hoka, S.; Ide, T.; Koitabashi, N.; Uchida, K.; Sumimoto, H.; Mori, Y.; Nishida, M., TRPC3-Nox2 complex mediates doxorubicin-induced myocardial atrophy. JCI insight 2017, 2, (15).

  1. While the histological differences between wild-type (WT) and Trpc3−/− mice hearts are mentioned, additional details on the histopathological findings would strengthen the study. Describing specific changes in cardiomyocyte morphology or interstitial fibrosis could provide a better understanding of the structural alterations observed in Trpc3−/− mice.

 Response: Thank you for your comment. To clearly delineate the differences in the RV between WT and TRPC3-/- mice under each condition, we have added the interstitial fibrosis assessment results using Masson's trichrome staining in Figure 2C. This additional data support the conclusion that the proper adaptive process of the RV in response to CIH is compromised in the absence of TRPC3.

  1. The lack of significant changes observed in pulmonary vessel thickness and the endothelin system in the lungs of CIH-exposed mice is intriguing. However, to better understand the role of TRPC3 in pulmonary hypertension, it would be valuable to further investigate other aspects of pulmonary vessel remodeling, such as endothelial dysfunction or smooth muscle cell proliferation.
  2. While the study focuses on RV functional parameters, it would be beneficial to also assess functional outcomes related to pulmonary hypertension, such as pulmonary arterial pressure measurements or right ventricular-pulmonary artery coupling. This would provide a more comprehensive understanding of the impact of TRPC3 deficiency on overall cardiovascular function.

 Response to 3 and 4: Thanks for the comments. In order to address this, we have reproduced the previous research findings at the time of planning this study. Specifically, we have added supplementary figure 3, which shows that RV hypertrophy induced by phenylephrine (PE) is not observed in TRPC3-/- mice. We have also added the relevance of the previous research results to the present study in the revised paper. Furthermore, we have described the vascular characteristics of the 129S strain mice used in this study, which have a different vascular stiffness index compared to the commonly used C57Bl/6j strain in pathological pulmonary vascular models. This additional description helps to supplementarily explain the absence of significant changes in lung vessel thickness and the endothelin system observed in our results.

“In previous experiments, we found that Trpc3−/− mice were resistant to RV dilation induced by high- dose Phenylephrine (PE) stimulation which was observed in WT mice [27]. PE, an α1 agonist, increased mesenteric arterial pressure and elevated pulmonary arterial pressure when administrated high doses, resulting in impaired RV systolic function [32] [33]. This phenomenon was confirmed again by our study (Supple. Fig. S3). The reduced RV dilation in Trpc3−/− mice during PE infusion can be attributed to decreased PA contraction due to the absence of TRPC3 similar contraction tendency induced by PE in the mesenteric artery and pulmonary artery can be assumed based on previous studies [37]. The absence of RV dilation in Trpc3−/− mice under PE infusion may be attributed to a decrease in pulmonary artery contraction due to the absence of TRPC3, suggesting that the loss of TRPC3's hypertrophic function in the myocardium was not problematic. On the other hand, under CIH conditions that induce PH, Trpc3−/− mice exhibited decreased RV function (Figure 1C), a significant decrease in RV parenchyma (Figure 2B), and increased fibrosis (Figure 2C) as well as increased expression of pathological markers (Figure 2D). The severe RV dilation observed in Trpc3−/− mice under CIH conditions might be a result of the absence of cellular hypertrophy necessary to resist the increase in PA pressure induced by CIH, even if there may have been some decrease in contractility due to the absence of TRPC3. There is a possibility that the specific hypoxic conditions of CIH, when met with the absence of TRPC3, further exacerbated the progression of appropriate compensatory processes.”

“It is very surprising that there was no significant difference in PA in this study. However, it is probably due to the vascular characteristics of the 129S strain mouse used in this study. In a previous report examining the major vascular stiffness indices of frequently used inbred mice, the blood pressure and heart rate of healthy animals were the same in all strains. However, a distinct difference in vascular stiffness indices was reported [39]. That is C57Bl/6J mice, which were mainly used in pulmonary artery-related studies, showed the lowest tensile stiffness and the highest acetylcholine-induced vascular relaxation, whereas 129S mice showed high tensile stiffness and the lowest acetylcholine-induced vascular relaxation. If pulmonary vasoconstriction and relaxation can be measured functionally in the CIH situation, it will be possible to determine whether there is no significant difference in pulmonary vascular changes based on these vascular characteristics. Studies investigating CIH utilizing the 129S mouse strain might more distinctly delineate the impact on cardiac functioning as opposed to vascular effects. The severe RV dysfunction observed in this study is likely attributable to an RV-specific mechanism in response to CIH.”

Reference

  1. Han, J. W.; Lee, Y. H.; Yoen, S.-I.; Abramowitz, J.; Birnbaumer, L.; Lee, M. G.; Kim, J. Y., Resistance to pathologic cardiac hypertrophy and reduced expression of CaV1.2 in Trpc3-depleted mice. Molecular and cellular biochemistry 2016, 421, (1-2), 55-65.
  2. Huang, J.-H.; He, G.-W.; Xue, H.-M.; Yao, X.-Q.; Liu, X.-C.; Underwood, M. J.; Yang, Q., TRPC3 channel contributes to nitric oxide release: significance during normoxia and hypoxia-reoxygenation. Cardiovascular research 2011, 91, (3), 472-482.
  3. Hirsch, L. J.; Rooney, M. W.; Wat, S. S.; Kleinmann, B.; Mathru, M., Norepinephrine and phenylephrine effects on right ventricular function in experimental canine pulmonary embolism. Chest 1991, 100, (3), 796-801.
  4. Steppan, J.; Jandu, S.; Wang, H.; Kang, S.; Savage, W.; Narayanan, R.; Nandakumar, K.; Santhanam, L., Commonly used mouse strains have distinct vascular properties. Hypertension Research 2020, 43, (11), 1175-1181.

  1. It would be helpful for the authors to acknowledge and discuss any limitations of the study, such as sample size, potential confounding factors, or the generalizability of findings to human OSA. Addressing these limitations will strengthen the study's conclusions and provide context for future research directions.

 Response: Thanks for your consideration to compensate the shortage of our study. We added the following paragraph to notify the limitation of our study.

 “Our study has limitations. First, we were unable to observe the progression of RV dysfunction in response to CIH, nor did we investigate the influence of the stimulus duration on the outcomes. Evaluating the interaction between PAP and the RV and PA is crucial to assess the severity and progression of PH, yet investigations into the functional outcomes associated with these factors were insufficient. Evaluating the interaction between PAP and the RV and PA is crucial to assess the severity and progression of PH. However, investigations into the functional outcomes associated with these factors need to be more comprehensive. Further studies are required to elucidate the molecular mechanisms through which TRPC3 contributes to RV adaptation with PH. The small sample size due to the low pregnancy rate of the Trpc3−/− mice we used, potential confounding factors, and the inability to directly generalize the findings from animals to human OSA constitute significant limitations of this study. Future research should incorporate such functional evaluations to comprehensively investigate the impacts of TRPC3 deficiency on PH and the associated functional outcomes.”

Minor comments

"We appreciate your comments for enhancing the readability for the readers. We have strived to structure the manuscript with shorter sentences throughout, and have made corrections for precise English grammar."

Round 2

Reviewer 1 Report

The authors sufficiently responded to the reviewers' concerns. 

Reviewer 2 Report

The authors have addressed my comments satisfactorily